# New estimates of genome size in Orthoptera and their evolutionary implications

Oliver Hawlitschek[1☯]*, David Sadílek[2☯], Lara-Sophie Dey[1], Katharina Buchholz[1,3], Sajad Noori[1,4], Inci Livia Baez[1,5], Timo Wehrt[1], Jason Brozio[6], Pavel Trávníček[7], Matthias Seidel[8], Martin Husemann[1]

1 Leibniz Institute for the Analysis of Biodiversity Change (LIB), Museum of Nature, Hamburg, Germany, 2 Institute of Medical Biochemistry and Laboratory Diagnostics, Centre of Oncocytogenomics, General University Hospital in Prague, Prague, Czech Republic, 3 Department of Biology, Norwegian University of Science and Technology (NTNU), Trondheim, Norway, 4 Staatliches Museum für Naturkunde Stuttgart, Stuttgart, Germany, 5 Leibniz Institute for the Analysis of Biodiversity Change (LIB), Museum Koenig, Bonn, Germany, 6 Zoologische Staatssammlung München (ZSM-SNSB), München, Germany, 7 Czech Academy of Sciences, Institute of Botany, Průhonice, Czech Republic, 8 Naturhistorisches Museum Wien, Wien, Austria

☯ These authors contributed equally to this work.
* oliver.hawlitschek@gmx.de

**Data Availability Statement:** All relevant data are within the paper and its Supporting Information

## Abstract

Animal genomes vary widely in size, and much of their architecture and content remains poorly understood. Even among related groups, such as orders of insects, genomes may vary in size by orders of magnitude–for reasons unknown. The largest known insect genomes were repeatedly found in Orthoptera, e.g., *Podisma pedestris* (1C = 16.93 pg), *Stethophyma grossum* (1C = 18.48 pg) and *Bryodemella holdereri* (1C = 18.64 pg). While all these species belong to the suborder of Caelifera, the ensiferan *Deracantha onos* (1C = 19.60 pg) was recently found to have the largest genome. Here, we present new genome size estimates of 50 further species of Ensifera (superfamilies Gryllidea, Tettigoniidea) and Caelifera (Acrididae, Tetrigidae) based on flow cytometric measurements. We found that *Bryodemella tuberculata* (Caelifera: Acrididae) has the so far largest measured genome of all insects with 1C = 21.96 pg (21.48 gBp). Species of Orthoptera with 2n = 16 and 2n = 22 chromosomes have significantly larger genomes than species with other chromosome counts. Gryllidea genomes vary between 1C = 0.95 and 2.88 pg, and Tetrigidae between 1C = 2.18 and 2.41, while the genomes of all other studied Orthoptera range in size from 1C = 1.37 to 21.96 pg. Reconstructing ancestral genome sizes based on a phylogenetic tree of mitochondrial genomic data, we found genome size values of >15.84 pg only for the nodes of *Bryodemella holdereri* / *B. tuberculata* and *Chrysochraon dispar* / *Euthystira brachyptera*. The predicted values of ancestral genome sizes are 6.19 pg for Orthoptera, 5.37 pg for Ensifera, and 7.28 pg for Caelifera. The reasons for the large genomes in Orthoptera remain largely unknown, but a duplication or polyploidization seems unlikely as chromosome numbers do not differ much. Sequence-based genomic studies may shed light on the underlying evolutionary mechanisms.

files. All measurement values will be provided to the Animal Genome Size Database.

**Funding:** Funding was provided by a grant from the Czech Academy of Sciences (RVO 67985939) to Pavel Trávníček. The funders had no role in study design, data collection and analysis, decision to publish, or preparation of the manuscript.

**Competing interests:** The authors have declared that no competing interests exist.

## Introduction

Despite the enormous advances in sequencing technology, much of the structures and functions of genomes remain poorly understood. One of these is the 'C-value enigma' or 'C-value paradox' [1], which relates to the issue that different species have highly variable contents of non-coding DNA despite similar amounts of coding DNA. Large amounts of non-coding DNA and, consequently, large genomes pose problems to genomic sequencing and genome assembly. Even genetic studies based on single-read sequencing (i.e., Sanger) may become complicated due to the high prevalence of paralogs [2]. Knowledge of at least the rough size of its genome is therefore a prerequisite for genomic studies on any organism. Unfortunately, the genome sizes of just relatively few species are known. The Animal Genome Size Database [3] holds records of 6,222 species as of 30 June 2022, representing less than 0.37% of the know 1.7 million species. Out of the more than 1 million described species of insects, the most diverse class of organisms, only 1,164 species have a total of 1,345 recordsuppls in the Animal Genome Size Database. While some species with small genome sizes are well-known as model organisms, e.g., *Drosophila melanogaster* with 1C = 0.18 pg [4] and *Tenebrio molitor* with 1C = 0.52 pg [5], many have much larger genomes. One order with exceptionally large genomes is Orthoptera.

For several years, grasshoppers of the family Acrididae have held the records for the largest insect genomes. These were *Podisma pedestris* (1C = 16.93 pg; [6]), *Bryodemella holdereri* (1C = 18.64 pg; [7]) and *Stethophyma grossum* (1C = 18.48 pg; [8]). Satellite DNAs and transposable elements have been suggested as potential explanations for the large sizes [9, 10]. Complete genome duplications may be less likely, as there is a lack of correlation of chromosome number and genome size. Despite their mostly higher chromosome numbers, ensiferans typically have smaller genomes than caeliferans [11]. Remarkably in this context, the most recent record holder for genome size in Orthoptera is the ensiferan *Deracantha onos* (1C = 19.60 pg; [12]). These studies, as reviewed by Gregory [3], show that genome sizes vary widely in grasshoppers and probably also all other groups of Orthoptera, warranting further investigation.

To obtain a better understanding of genome size variation in Orthoptera and its underlying evolutionary mechanisms, we generated estimates of genome size for 50 species of Orthoptera and used a mitogenomic phylogeny to track genome size across the evolution of the group. The main goals of this study were: 1) To provide measurements of the genome size of further species of Orthoptera and thus improve our knowledge on the range and variation of this character. 2) To track the evolution of genome size along the phylogenetic tree of Orthoptera. 3) To compare genome size data with chromosome numbers and, in the light of the XX/X0 sex determination system, discuss their implications for future studies.

## Materials and methods

### Sampling

We collected specimens at eight sites across Germany and one in Austria: Meadows around Motzen, Brandenburg, 52.2013 13.5892; Meadows around Pevestorf, Lower Saxony, 53.0610 11.4578; Railway parking area Munich Allach (Rangierbahnhof München Nord), Bavaria, 48.1902 11.5318; Eglinger Filz bogs near Wolfratshausen, Bavaria, 47.9016 11.5060; Upper Isar river near Sylvenstein, Bavaria, 47.5631 11.4746; Fröttmaninger Heide meadows North of Munich, Bavaria, 48.2128 11.6153; Alpiner Steig rock outcrops near Nittendorf, Bavaria, 49.0037 11.9680; Sudelfeld alpine meadows, Bavaria, 47.6835 12.0371; Rofanspitze alpine meadows, Tyrol, Austria, 47.4531 11.7892. We furthermore obtained specimens of some species that do not naturally occur in Germany from the pet trade. The identification of freshly

collected species was based on morphological and bioacoustic characters [13, 14]. We follow the nomenclature of Cigliano et al. [15]. Voucher specimens were deposited in the Zoological Museum Hamburg (ZMH), part of the Leibniz Institute for the Analysis of Biodiversity Change (LIB) under the accession ZMH 2019/21. A detailed list of all specimens and samples with individual accession numbers is given in S1 Table.

The Government of Upper Bavaria authorized handling, capture, and killing of the insect specimens used in this study with a permit issued on 15 July 2019. Insects were anesthetized and euthanized using CO2.

## Genome size measurements

We measured the nuclear DNA content (2C) of samples using the flow cytometry method (FCM) as described in Sadilek et al. [16, 17] (see also [8]). For every sample, we extracted the muscle tissue of one hind femur and homogenized it with a leaf of the internal plant standard *Pisum sativum* L. "Ctirad" (Fabaceae) with 2C = 9.09 pg [18, 19] in 500 µl of Otto buffer I at 4°C. A plant standard was used to ensure comparability with previously conducted measurements of the same facility (8). We then filtered the cell suspension through a 42 µm nylon mesh and split it in two halves. One half was stained with 1,000 µl DAPI solution (1 ml of Otto II buffer (0.4 M $Na_2HPO_4 \cdot 12\ H_2O$) supplemented by AT-selective fluorescent dye DAPI (4',6-diamino-2-phenylindol) and 2-mercaptoethanol in final concentrations of 4 µg/ml and 2µl/ml, respectively) and the second half with 1,000 µl propidium iodide (PI) solution (1 ml of Otto II buffer (0.4 M $Na_2HPO_4$ * 12 $H_2O$ supplemented by intercalating fluorescent dye PI, RNase and 2-mercaptoethanol in final concentrations of 50 µg/ml, 50 µg/ml and 2µl/ml, respectively) for several minutes. We conducted the analysis of the DAPI-stained sample in a Partec CyFlow instrument with an UV LED chip and the analysis of the PI-stained sample in a Partec SL instrument with a green solid-state laser (Cobolt Samba, 532 nm, 100 mW; Partec GmbH, Münster, Germany). 3,500 to 5,000 particles were recorded in each FCM analysis. We analyzed the output data with the Partec FloMax v. 2.52 software (Partec GmbH, Münster, Germany). Sample genome size was calculated as ratio of known standard genome size to measured peak. Median coefficients of variation are 2.31 for DAPI and 3.81 for PI. All measurements and analyses were conducted at the Institute of Botany of Czech Academy of Sciences, Prague.

Combined DAPI and PI measurement results of the same specimen express the AT/GC ratio of the genome of the species, the GC content (e.g.: [16, 17, 20]). The GC content of *P. sativum* is 38.50% [21] and the GC content of the analyzed samples was calculated with a Microsoft Excel macro [20]. Measurements in pg were converted to base pairs *$10^9$ (Gbp) using the formula 1 pg = 0.978 Gbp [22].

## Analyses of genome size data

We assembled a dataset of newly measured species and Orthoptera genome size measurements from previous studies, based on the Animal Genome Size Database [3] complemented with further recent studies. We then plotted the male genome sizes against the number of chromosomes [11] for all species for which both values were available; we used male genome size because more male measurements are available from the literature combined with our new data. We tested for statistical significance using a Kruskal-Wallis test and pairwise Mann-Whitney tests (Bonferroni corrected) of all chromosome numbers with more than one record in PAST 4.03 [23]. Finally, we used our data to calculate the difference between mean male and female genome size for each species where both sexes were available and tested for correlation of size differences between sexes and male genome size with Pearson's r.

We also tested for correlation between genome size and GC content among our new measurements, independently for female and male specimens. We then checked if GC content was generally different between females and males using a t-test. As this test yielded non-significant results (see Results), we used an ANOVA and pairwise Mann-Whitney tests (Bonferroni corrected) to test for differences in GC content between families.

### The evolution of genome size in Orthoptera

In order to track the development of genome size along the evolutionary history of Orthoptera, we plotted the known genome sizes on a phylogenetic tree. We assembled a dataset of complete and partial mitochondrial genomes for tree reconstruction with our new measurements combined with data from GenBank and BOLD [24] (S1 Table). Out of the 146 species with known genome sizes, we found complete mitochondrial genomes available for 86 individuals belonging to 70 species. Under the rationale that all species included in the dataset should at least be represented in the mitochondrial genes COI, CytB, COII or ND5, we added 49 further species for which at least one of these additional mitochondrial markers was available. We aligned the dataset using MUSCLE [25] integrated in Geneious v.10.0.9 [26] and KALIGN [27]. Since many regions of the mitogenome were represented in only relatively few specimens, we reduced the dataset to the genes Cytochrome C Oxidase I and II, Cytochrome B, and NADH Dehydrogenase 5.

We then reconstructed a Maximum Likelihood tree using the IQtree web server [28, 29] with automatic substitution model selection, 1,000 Bootstrap alignments, and 1,000 iterations under a minimum correlation coefficient of 0.99, treating all mitochondrial genes as one partition. The single branch test was performed after 1,000 replicates, a perturbation strength of 0.5, and an internal IQ-Tree stopping rule of 100. Based on this tree, we reconstructed ancestral states of genome size in the R v.3.6.3. environment [30] using the packages 'phytools' [31] and 'ape' [32]. The ape package implements non-parametric rate smoothing and does not require ultrametrization of the tree. The R code is available under https://github.com/laradey/Genome_size_Orthoptera.git. In species for which more than one measurement was available, we used the mean value.

## Results

### Genome size variation

We provide newly measured genome size data for 103 individuals assigned to 50 species of the families Acrididae, Tetrigidae, Gryllidae, and Tettigoniidae, of which 38 species were measured for the first time (Table 1). The largest genome measured in this study is 1C = 21.96 pg (21.48 Gbp) and belongs to the speckled buzzing grasshopper *Bryodemella tuberculata* (Acrididae: Oedipodinae). The second largest genome belongs to *Chrysochraon dispar* (Acrididae: Gomphocerinae; 1C = 19.43 pg, 19.00 Gbp), followed by *Stethophyma grossum* (Acrididae: Oedipodinae; 1C = 18.51 pg, 18.10 Gbp).

Table 2 compares genome sizes of families and subfamilies of all Orthoptera and shows that all subgroups studied here have large genome sizes compared to insect model species [3]. Within Orthoptera, Acrididae have exceptionally large genomes and representatives of the subfamilies Oedipodinae (max. 1C = 21.96 pg), Gomphocerinae (max. 1C = 18.76 pg), and Melanoplinae (max. 1C = 16.93) have larger genomes than those of other subfamilies of Acrididae (1C = 7.50–13.96 pg).

Plotting genome size vs. chromosome numbers (Fig 1) suggests correlations for some chromosome counts. The Kruskal-Wallis test of correlation between genome size and chromosome number was highly significant with p = $1.196E^{-07}$. Mann-Whitney tests were significant

**Table 1. A list of genomic and cytogenetical data on all 50 species measured for this study.**

| Species | N | 1C F [pg] | 1C M [pg] | GC [%] | 2n |
|---|---|---|---|---|---|
| **Acrididae** | | | | | |
| *Bryodemella tuberculata* | 2F | 21.92 | - | 42.05 | 22+X0 |
| *Calliptamus italicus* | 2F,1M | 11.68 | 10.91 | 42.66 | 22+X0* |
| *Euthystira brachyptera* | 2F | 17.95 | - | 41.51 | 16+X0 |
| *Gomphocerippus rufus* | 2F | 13.18 | - | 41.53 | 16+X0 |
| *Chorthippus albomarginatus* | 1F,1M | 11.88 | 11.79 | 41.20 | 16+X0 |
| *Chorthippus apricarius* | 2F,1M | 12.52 | 11.92 | 40.84 | 16+X0* |
| *Chorthippus biguttulus* | 1M | - | 10.99 | 41.80 | 16+X0 |
| *Chorthippus brunneus* | 1M | | 10.47 | 41.17 | 16+X0 |
| *Chorthippus dorsatus* | 2F,1M | 12.59 | 12.80 | 41.39 | 16+X0 |
| *Chorthippus mollis* | 1M | - | 11.58 | 41.50 | 16+X0* |
| *Chorthippus pullus* | 1F | 13.44 | - | 41.32 | 16+X0* |
| *Chorthippus vagans* | 2F | 11.11 | - | 41.30 | 16+X0 |
| *Chrysochraon dispar* | 1F,1M | 19.43 | 18.76 | 41.43 | 16+X0 |
| *Locusta migratoria* | 1M | - | 7.62 | 41.52 | 22+X0 |
| *Myrmeleotettix maculatus* | 2F | 11.83 | - | 41.40 | 16+X0 |
| *Oedipoda caerulescens* | 2F | 14.13 | - | 42.13 | 22+X0 |
| *Omocestus haemorrhoidalis* | 1F,1M | 12.83 | 12.14 | 41.00 | 16+X0 |
| *Omocestus viridulus* | 1F,1M | 14.03 | 13.28 | 41.11 | 16+X0 |
| *Pseudochorthippus montanus* | 1F,1M | 13.12 | 12.42 | 41.22 | 16+X0* |
| *Pseudochorthippus parallelus* | 2F,1M | 13.14 | 12.67 | 41.83 | 16+X0 |
| *Psophus stridulus* | 1M | - | 16.44 | 41.95 | 22+X0 |
| *Schistocerca gregaria* | 1F,1M | 10.68 | 10.36 | 43.40 | 22+X0 |
| *Stenobothrus lineatus* | 1F,1M | 14.00 | 13.63 | 41.25 | 16+X0 |
| *Stenobothrus nigromaculatus* | 1F,1M | 13.18 | 12.48 | 41.05 | 16+X0* |
| *Stenobothrus stigmaticus* | 1F,1M | 11.91 | 11.21 | 41.24 | 16+X0* |
| *Stethophyma grossum* | 1F | 18.51 | - | 42.56 | 22+X0 |
| **Gryllidae** | | | | | |
| *Acheta domesticus* | 1F,1M | 2.88 | 2.63 | 39.13 | 10+X0 |
| *Gryllus assimilis* | 1F,1M | 2.24 | 2.09 | 38.58 | 28+X0 |
| *Gryllus bimaculatus* | 1F,1M | 2.22 | 1.98 | 38.83 | 28+X0 |
| *Gryllus campestris* | 1F,1M | 2.23 | 2.08 | 39.02 | 28+X0 |
| *Nemobius sylvestris* | 1F | 2.56 | - | 36.41 | 16+X0 |
| *Oecanthus pellucens* | 1F,1M | 1.44 | 1.37 | 39.97 | 18+XY |
| **Tetrigidae** | | | | | |
| *Tetrix subulata* | 1M | - | 2.22 | 35.62 | 12+X0 |
| *Tetrix tuerki* | 2F | 2.37 | - | 36.08 | 12+X0* |
| *Tetrix undulata* | 1F,1M | 2.36 | 2.18 | 35.84 | 12+X0 |
| **Tettigoniidae** | | | | | |
| *Bicolorana bicolor* | 1F,1M | 8.05 | 6.99 | 39.68 | 30+X0 |
| *Conocephalus dorsalis* | 1M | - | 3.52 | 39.32 | 32+X0 |
| *Conocephalus fuscus* | 1F,1M | 4.42 | 3.79 | 39.57 | 32+X0* |
| *Decticus verrucivorus* | 2F,2M | 8.21 | 7.34 | 41.01 | 30+X0 |
| *Leptophyes punctatissima* | 1F,3M | 7.98 | 6.81 | 41.03 | 30+X0 |
| *Meconema meridionale* | 1F,1M | 10.69 | 9.90 | 41.23 | 26+X0* |
| *Meconema thalassinum* | 2F | 12.72 | - | 40.85 | 26+X0 |
| *Metrioptera brachyptera* | 1F,1M | 8.78 | 7.97 | 39.93 | 30+X0 |

*(Continued)*

**Table 1.** (Continued)

| Species | N | 1C F [pg] | 1C M [pg] | GC [%] | 2n |
|---|---|---|---|---|---|
| *Phaneroptera falcata* | 1F,1M | 7.25 | 6.08 | 38.78 | 26+X0 |
| *Pholidoptera griseoaptera* | 2F,2M | 7.11 | 6.30 | 40.73 | 30+X0 |
| *Pholidoptera littoralis* | 1F | 7.69 | - | 40.20 | 30+X0* |
| *Platycleis albopunctata* | 2F,2M | 6.54 | 5.74 | 39.51 | 30+X0* |
| *Roeseliana roeselii* | 1F,3M | 8.30 | 7.70 | 40.30 | 30+X0 |
| *Tettigonia cantans* | 1F,1M | 7.16 | 6.34 | 40.89 | 28+X0 |
| *Tettigonia viridissima* | 2M | - | 5.69 | 42.88 | 28+X0 |

See S1 Table for individual details. **N** = number of females and males analyzed, **1C** = average haploid genome size for females and males, **GC** = average content of GC basis in the genome of the species, **2n** = male diploid chromosome number, * = chromosome number available only for another species of the same genera.

($p \leq 0.05$) or highly significant ($p \leq 0.001$, in the case of 2n = 22 and 2n = 32) for all pairwise comparisons of chromosome numbers of 2n = 16 with any other chromosome numbers except 2n = 14 and 2n = 18, and for most pairwise comparisons of 2n = 22 (Table 3).

The genome size differences between sexes are given in Table 4, including 83 species. The highest genome size differences were detected within four species of Acrididae (all 2n = 16 +X0)–with maximum of 2.52 pg in *Gomphocerippus rufus* (1C (male) = 10.66 pg), followed by *Chorthippus vagans* (2.43 / 8.68 pg), *Pseudochorthippus parallelus* (2.25 / 10.89 pg), and *Schistocerca gregaria* (2.13 / 8.55 pg). The next largest differences in the genome size were found in Tettigoniidae–with maximum of 1.75 pg in *Deracantha onos* (1C (male) = 17.39 pg, 2n = 28 +X0). Expectably, we found a significant correlation between larger genome size differences between sexes and larger male genome sizes ($p = 0.005^*$). The difference in genome size between female and male (XX/X0) can be interpreted also as the size of the sex chromosome X. Negative size differences, i.e., larger genome sizes of males than of females, were detected in *Myrmeleotettix maculatus* (Acrididae; 1C (male) = 12.14 pg, 2n = 16+X0) with -0.31 pg and *Chorthippus dorsatus* (Acrididae; 1C (male) = 12.80 pg, 2n = 16+X0) with -0.21 pg.

We found genome size and GC content highly correlated in both females ($p = 5.924E^{-12}$) and males ($p = 9.597E^{-06}$), with smaller genomes mostly having lower GC content. The GC content range was 35.86%-43.17% (N = 57, mean = 40.63±1.61%) for females and 35.58%-44.21% (N = 46, mean = 40.56±1.59%) for males. Conspecific female and male GC content did not differ significantly ($p = 0.826$). The ANOVA test of differences of GC content among the families Acrididae, Tetrigidae, Gryllidae, and Tettigoniidae was highly significant ($p = 5.965E^{-29}$), and all pairwise Mann-Whitney tests (Bonferroni corrected) among these families were significant or highly significant, except between Tetrigidae and Gryllidae (Table 5).

## Evolutionary analysis

The phylogenetic tree based on mitochondrial genomes generated with IQtree is given as partial trees of Ensifera (Fig 2) and Caelifera (Fig 3) (see also S1 File for the complete tree). The tree is overall well supported with bootstrap values of >95. Caelifera and Ensifera were found monophyletic, as are Gryllidea and Tettigoniidea within Ensifera. Caelifera is only represented by Acrididea, since no members of Tridactylidea were included. The genus *Meconema* is found as the sister group to all other Tettigoniidea, making Tettigoniidae paraphyletic with respect to all other families of this infraorder. All subfamilies of Tettigoniidae included are retrieved as monophyletic. The only genera of this group found paraphyletic are *Ruspolia* (with respect to *Neoconocephalus*) and *Metrioptera* (with respect to *Bicolorana*). Within Caelifera, both Tetrigidae and Acrididae, as well as all subfamilies of Acrididae included by us, were

**Table 2. An overview comparison of approximate genome sizes (1C) of Orthoptera families and subfamilies analyzed so far.**

| | Species | MIN | MAX | Average | male 2n |
|---|---|---|---|---|---|
| **CAELIFERA** | **80** | **2.18** | **21.92** | **5.45** | **13,15,17,19,21,23** |
| **Acrididae** | **76** | **5.83** | **21.92** | **10.17** | **17,19,21,23** |
| Acridinae | 5 | 7.50 | 12.55 | 10.43 | 23 |
| Calliptaminae | 3 | 9.64 | 11.68 | 10.41 | 23 |
| Catantopinae | 3 | 8.49 | 10.73 | 9.90 | 23 |
| Cyrtacanthacridinae | 4 | 8.63 | 10.68 | 9.28 | 23 |
| Eyprepocnemidinae | 3 | 6.34 | 9.70 | 7.42 | 23 |
| Gomphocerinae | 32 | 8.58 | 19.43 | 12.39 | 17,23 |
| Melanoplinae | 7 | 5.83 | 16.93 | 8.98 | 21,23 |
| Oedipodinae | 16 | 5.99 | 21.92 | 11.04 | 23 |
| Pyrgomorphinae | 1 | 7.55 | 8.21 | 7.88 | 19 |
| Thrinchinae | 2 | 13.51 | 14.45 | 13.96 | 19 |
| **Morabidae** | **1** | **3.75** | **4.00** | **3.88** | **15** |
| Morabinae | 1 | 3.75 | 4.00 | 3.88 | 15 |
| **Tetrigidae** | **3** | **2.18** | **2.41** | **2.30** | **13** |
| Tetriginae | 3 | 2.18 | 2.41 | 2.30 | 13 |
| **ENSIFERA** | **63** | **0.95** | **19.14** | **4.92** | **11,13,15,17,20,21,23, 27,29,31,33,35,37** |
| **Anostostomatidae** | **2** | **5.40** | **6.53** | **5.97** | **15** |
| Deinacridinae | 2 | 5.40 | 6.53 | 5.97 | 15 |
| **Gryllacrididae** | **1** | **9.45** | **9.45** | **9.45** | **unknown** |
| Gryllacridinae | 1 | 9.45 | 9.45 | 9.45 | unknown |
| **Gryllidae** | **14** | **0.95** | **2.88** | **2.12** | **11,13,17,20,21,27,29** |
| Eneopterinae | 1 | 2.09 | 2.35 | 2.22 | unknown |
| Gryllinae | 8 | 1.98 | 2.88 | 2.29 | 11,13,21,27,29 |
| Nemobiinae | 1 | 2.56 | 2.56 | 2.56 | 17 |
| Oecanthinae | 3 | 0.95 | 1.71 | 1.31 | 20 |
| Podoscirtinae | 1 | 2.23 | 2.23 | 2.23 | unknown |
| **Gryllotalpidae** | **2** | **3.41** | **4.21** | **3.81** | **unknown** |
| Gryllotalpinae | 2 | 3.41 | 4.21 | 3.81 | unknown |
| **Mogoplistidae** | **1** | **3.08** | **3.48** | **3.28** | **unknown** |
| Mogoplistinae | 1 | 3.08 | 3.48 | 3.28 | unknown |
| **Rhaphidophoridae** | **3** | **1.55** | **9.55** | **5.44** | **29,35,37** |
| Aemodogryllinae | 1 | 5.15 | 5.48 | 5.32 | 29 |
| Ceuthophilinae | 2 | 1.55 | 9.55 | 5.55 | 35,37 |
| **Tettigoniidae** | **38** | **2.65** | **19.14** | **9.63** | **21,23,27,29,31,33** |
| Bradyporinae | 2 | 12.71 | 19.14 | 15.80 | 29,31 |
| Conocephalinae | 9 | 2.65 | 10.05 | 6.09 | 21,23,33 |
| Hexacentrinae | 1 | 12.80 | 14.01 | 13.41 | 31,33 |
| Meconematinae | 3 | 4.38 | 12.44 | 8.96 | 27 |
| Mecopodinae | 1 | 13.45 | 14.58 | 14.02 | 27,29 |
| Phaneropterinae | 7 | 5.09 | 10.58 | 7.10 | 27,29,31 |
| Pseudophyllinae | 2 | 3.47 | 5.91 | 4.69 | unknown |
| Tettigoniinae | 13 | 5.34 | 8.78 | 6.97 | 29,31 |
| **Tridactylidae** | **1** | **2,63** | **2,63** | **2,63** | **11,13** |
| **Trigonidiidae** | **1** | **1.93** | **1.93** | **1.93** | **15** |

*(Continued)*

**Table 2.** (Continued)

| | Species | MIN | MAX | Average | male 2n |
|---|---|---|---|---|---|
| Trigonidiinae | 1 | 1.93 | 1.93 | 1.93 | 15 |

Males and females of each taxon are analyzed together. See S1 Table for individual details. Data for Deinacridinae are given in relative genome size as the samples were measured with DAPI stain. DAPI measurements are influenced by AT/GC ratio in the genome and are considered less accurate than the PI stain used in all other FCM values listed here.

monophyletic. The genus *Chorthippus* is found polyphyletic: *Chorthippus pullus* is placed in a group containing *Myrmeleotettix*, *Omocestus*, and *Stenobothrus*. The remainder of *Chorthippus* is paraphyletic with respect to *Gomphocerippus*, *Gomphocerus*, and *Stauroderus*.

Plotting genome sizes on the tree shows consistent values of mean genome size in different clades of the Orthoptera tree (Fig 4). The sizes of all Gryllidea genomes are between 1C = 0.91 pg (*Oecanthus sinensis*) and 2.88 pg (*Acheta domesticus*), with the exceptions of *Neoscapteriscus borellii* (3.41), *Ornebius kanetataki* (3.28) and *Gryllotalpa orientalis* (4.21). The values in Tetrigidae range between 2.22 and 2.36. On the other hand, all members of Tettigoniidea and Acrididae show values between 4.01 and 14.15. The exceptions are *Hadenoecus subterraneus* (1.55) at the lower end and *Stauroderus scalaris* (15.04 / 16.34), *Psophus stridulus* (16.44), *Podisma pedestris* (16.93), *Eusthystira brachyptera* (17.94), *Stethophyma grossum* (18.11), *Deracantha onos* (18.26), *Bryodemella holdereri* (18.41), *Chrysochraon dispar* (19.09), and *B. tuberculata* (21.92).

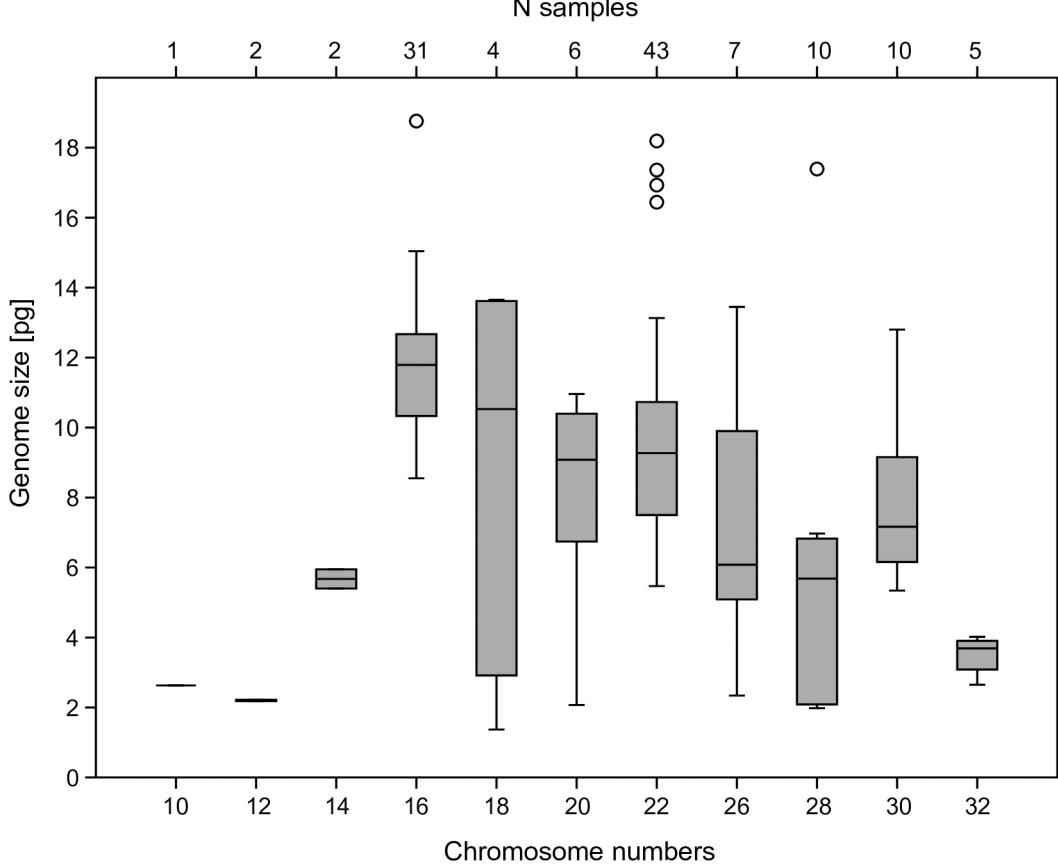

**Fig 1. Box plot of number of chromosomes (2n+X0) in male Orthoptera vs. genome size (1C [pg]).**

**Table 3. Mann-Whitney tests (Bonferroni corrected) for all pairwise comparisons of chromosome numbers of 2n = 16 and 2n = 22 with any other chromosome numbers except 2n = 18 and 2n = 20.**

|    | 12 | 14 | 16 | 18 | 20 | 22 | 26 | 28 | 30 |
|----|----|----|----|----|----|----|----|----|----|
| 14 | 0.2453 | | | | | | | | |
| 16 | 0.0214* | 0.0214 | | | | | | | |
| 18 | 0.4875 | 0.4875 | 0.8155 | | | | | | |
| 20 | 0.2433 | 0.2433 | 0.0061* | 0.9151 | | | | | |
| 22 | 0.0192* | 0.0223* | 0.0001** | 0.9241 | 0.772 | | | | |
| 26 | 0.057 | 0.6605 | 0.0084* | 0.5083 | 0.5203 | 0.0962 | | | |
| 28 | 0.4521 | 0.9145 | 0.0002* | 0.358 | 0.0927 | 0.0006** | 0.3539 | | |
| 30 | 0.0413* | 0.1626 | 0.0013* | 0.4367 | 0.3028 | 0.0577 | 0.6605 | 0.0452* | |
| 32 | 0.0814 | 0.0814 | 0.0004** | 0.2703 | 0.0828 | 0.0003** | 0.0513 | 0.2446 | 0.0027* |

* = significant ($p \leq 0.05$)

** = highly significant ($p \leq 0.001$). Kruskal-Wallis: $Chi^2$ 49.76, $p = 1.196E^{-7}$

The ancestral state reconstruction found genome size values of >15.84 pg only for the nodes of *Bryodemella holdereri / B. tuberculata* and *Chrysochraon dispar / Euthystira brachyptera* (S1 File). The predicted genome size values are 1C = 6.19 pg for Orthoptera, 5.37 pg for Ensifera, and 7.28 pg for Caelifera.

## Discussion

The diversity in genome size is an important parameter in organismal diversity research, yet it remains poorly studied. This is particularly true for insects, where genome size is known for less than 0.5% of the described species [3]. So far, the largest genomes of all insects have been detected among members of Orthoptera, exceeding the human genome by the factor seven. Nevertheless, data is available for only a small selection of species. The variation in measurement methods and standards also somewhat limits the comparability of genome sizes from different studies and databases. However, the commonly observed congruence between values from different sources (S1 File) suggests to us that joint analyses are valid. We contributed measurement of 50 species, 38 measured for the first time, and added these to the known set of species. The variation in genome sizes was even larger than expected but does not follow a clear pattern.

### Genome size variation in Orthoptera

The newly measured size of 1C = 21.96 pg of the genome *Bryodemella tuberculata* surpasses the previous records held by 1C = 19.60 pg in *Deracantha onos* [12], 1C = 18.48 pg in *B. holdereri* [7], 1C = 18.48 pg in *Stethophyma grossum* [8], and 1C = 16.93 pg in *Podisma pedestris* [6]. We are not aware of any insect with a larger genome published in the meantime. Therefore, *B. tuberculata* holds the record for the largest known insect genome size. The genome size of *Chrysochraon dispar* (1C (female) = 19.43) also surpasses that of all previously known genomes of Caelifera. Furthermore, the size of the largest genome measured of *Stethophyma grossum* slightly exceeds the value Husemann et al. [8] found in that species. This intraspecific or measurement variation is within the ranges detected in other similar studies [7, 12].

Schielzeth et al. [33] provided a measurement of the genome size of *Chorthippus biguttulus* (Acrididae) of 1C = 236.05 pg, which would exceed our measurement of *B. tuberculata* by an order of magnitude. We measured the genome size of *Ch. biguttulus* as 1C = 10.99 pg, which is in line with the measurements of Shah et al. [9] (1C = 9.31 pg) and Husemann et al. [8]

**Table 4. The genome size difference [pg] between the male and female measured by FCM, given for each species of completed dataset if data for both sexes were available (species from a single study was preferred).**

| Acrididae | | | |
|---|---|---|---|
| *Acrida cinerea* | 0.60 | *Melanoplus differentialis* | 0.47 |
| *Atractomorpha sinensis* | 0.66 | *Myrmeleotettix maculatus* | -0.31** |
| *Bryodemella holdereri* | 0.45 | *Oedaleus asiaticus* | 0.59 |
| *Calliptamus abbreviatus* | 0.39 | *Oedaleus infernalis* | 0.56 |
| *Calliptamus barbarus* | 0.41 | *Omocestus haemorrhoidalis* | 0.69 |
| *Calliptamus italicus* | 0.77 | *Omocestus viridulus* | 0.87 |
| *Campylacantha olivacea* | 0.83 | *Pararcyptera microptera meridionalis* | 0.75 |
| *Chorthippus albomarginatus* | 0.09 | *Pedopodisma tsinlingensis* | 0.88 |
| *Chorthippus apricarius* | 0.60 | *Pseudochorthippus montanus* | 0.70 |
| *Chorthippus biguttulus* | 0.32* | *Pseudochorthippus parallelus* | 2.25 |
| *Chorthippus dorsatus* | -0.21 | *Schistocerca gregaria* | 2.13* |
| *Chorthippus vagans* | 2.43** | *Shirakiacris shirakii* | 0.49 |
| *Chrysochraon dispar* | 0.67 | *Sinopodisma qinlingensis* | 0.39 |
| *Epacromius coerulipes* | 0.41 | *Sphingonotus caerulans* | 0.76 |
| *Euchorthippus unicolor* | 0.87 | *Stenobothrus lineatus* | 0.37 |
| *Filchnerella rubimargina* | 0.70 | *Stenobothrus nigromaculatus* | 0.70 |
| *Fruhstorferiola huayinensis* | 0.32 | *Stenobothrus stigmaticus* | 0.70 |
| *Gomphocerippus rufus* | 2.52* | *Stethophyma grossum* | 1.15* |
| *Haplotropis brunneriana* | 0.80 | *Trilophidia annulata* | 0.69 |
| *Locusta migratoria* | 0.97** | | |
| **Anostostomatidae** | | | |
| *Hemideina crassidens* | 0.61 | *Hemideina thoracica* | 0.58 |
| **Gryllidae** | | | |
| *Acheta domesticus* | 0.25 | *Oecanthus pellucens* | 0.07 |
| *Gryllodes sigillatus* | 0.20 | *Oecanthus sinensis* | 0.13 |
| *Gryllus assimilis* | 0.15 | *Teleogryllus emma* | 0.27 |
| *Gryllus bimaculatus* | 0.24 | *Xenogryllus marmoratus* | 0.26 |
| *Gryllus campestris* | 0.15 | | |
| **Mogoplistidae** | | **Rhaphidophoridae** | |
| *Ornebius kanetataki* | 0.40 | *Diestrammena* sp. | 0.33 |
| **Tetrigidae** | | | |
| *Tetrix undulata* | 0.18 | | |
| **Tettigoniidae** | | | |
| *Atlanticus sinensis* | 0.35 | *Metrioptera bonneti* | 0.56 |
| *Bicolorana bicolor* | 1.06 | *Metrioptera brachyptera* | 0.81 |
| *Conocephalus fuscus* | 0.63 | *Microconema clavata* | 0.34 |
| *Conocephalus gladiatus* | 0.52 | *Neoconocephalus triops* | 0.64 |
| *Conocephalus maculatus* | 0.30 | *Phaneroptera falcata* | 1.17 |
| *Conocephalus* sp. | 0.38 | *Phaneroptera gracilis* | 1.02 |
| *Decticus verrucivorus* | 0.87 | *Pholidoptera griseoaptera* | 0.81 |
| *Deracantha onos* | 1.75 | *Platycleis albopunctata* | 0.81 |
| *Ducetia japonica* | 0.87 | *Pseudorhynchus crassiceps* | 1.28 |
| *Elimaea berezovskii* | 0.74 | *Roeseliana roeselii* | 0.59 |
| *Hexacentrus unicolor* | 1.21 | *Ruidocollaris sinensis* | 0.97 |
| *Kuwayamaea brachyptera* | 1.53 | *Ruspolia dubia* | 0.61 |
| *Leptophyes punctatissima* | 1.17 | *Ruspolia lineosa* | 0.74 |

*(Continued)*

**Table 4.** (Continued)

| | | | |
|---|---|---|---|
| *Meconema meridionale* | 0.79 | *Tettigonia cantans* | 0.82 |
| *Mecopoda elongata* | 1.13 | *Zichya tenggerensis* | 1.24 |

See S1 Table for individual details. Data for Anostostomatidae are given in relative genome size due to the samples were measured with DAPI stain which is AT specific.

\* = difference calculated between specimens from diverse studies, both measured by FCM method

\*\* = difference calculated between specimens from diverse studies, one measured by FCM and second by FD method

(1C = 11.31 pg). This suggests that, as commented by Camacho [34], the measurement of Schielzeth et al. [33] was indeed unreliable and does not represent a true value.

Within Acrididae, the largest genomes belong to representatives of the subfamilies Oedipodinae (maximum: *Bryodemella tuberculata*, 2n = 22+XX, 1C = 21.96 pg), Gomphocerinae (*Chrysochraon dispar*, 2n = 16+XX, 1C = 19.43 pg), and Melanoplinae (*Podisma pedestris*, 2n = 22+X0, 1C = 16.93 pg). Fig 1 shows that species with male chromosome counts of 2n = 16 +X0, followed by 2n = 22+X0, have the largest genomes [11]. All these species belong to the family Acrididae. The representatives of other families of Caelifera, Tetrigidae, and Morabidae have far smaller recorded genomes. Overall, the genomes measured in Orthoptera so far span a large size range from less than 1 GB in some crickets to more than 20 GB as measured here for Oedipodinae. This suggests complex evolutionary processes underlying the evolution of genomes in Orthoptera, which will have to be explored in the future. So far, few Orthoptera genomes have been sequenced [35] owing to their large size, but comparative genomic analyses across genomes of different sizes will be necessary to understand the genome gigantism in this group.

Our study adds to other recent works [7, 8, 12], in providing new records of the largest genome size in Orthoptera and at the same time in all insects by studying just a comparatively limited number of species. We consider it very likely that future studies will discover even larger genomes in Orthoptera or among members of another insect order.

## The evolution of genome size

In order to study the evolution of genome size in Orthoptera, we plotted the known measurements on a phylogenetic tree based on mitochondrial data. We selected mitochondrial data because it was available for a large number of species included in our genome size dataset (110 out of 146 = 75.3%). The tree obtained by Maximum Likelihood reconstruction is largely congruent with other trees available for Orthoptera so far [36–40]. However, it does not resolve the positions of Gryllotalpidae (*Gryllotalpa*), Mogoplistidae (*Ornebius*), and Trigoniidae (*Nemobius*), which have been placed as hierarchical sister groups to Gryllidae in other studies [40, 41]. We found Tettigoniidae paraphyletic with respect to a clade consisting of Anostostomatidae (*Hemideina*) + Rhaphidophoridae (*Ceuthophilus* + *Hadenoecus*) that is placed as sister

**Table 5. A comparison of GC content between genomes of the families Acrididae, Tetrigidae, Gryllidae, and Tettigoniidae.**

| Family | N | Max | Min | Mean | MV Tetrigidae | MV Gryllidae | MV Tettigoniidae |
|---|---|---|---|---|---|---|---|
| Acrididae | 49 | 43.62 | 40.61 | 41.58±0.64 | 0.016\* | 1.65E-06\*\* | 4.43E-09\*\* |
| Tetrigidae | 5 | 36.29 | 35.58 | 35.89±0.31 | | 0.131 | 0.002\* |
| Gryllidae | 11 | 40.21 | 36.41 | 38.86±0.98 | | | 4.17E-04\*\* |
| Tettigoniidae | 38 | 44.21 | 38.7 | 40.46±1.00 | | | |

Mean is given ± standard deviation. Mann-Whitney (MV) tests (Bonferroni corrected) are highlighted as \* = significant ($p \leq 0.05$) or \*\* = highly significant ($p \leq 0.001$).

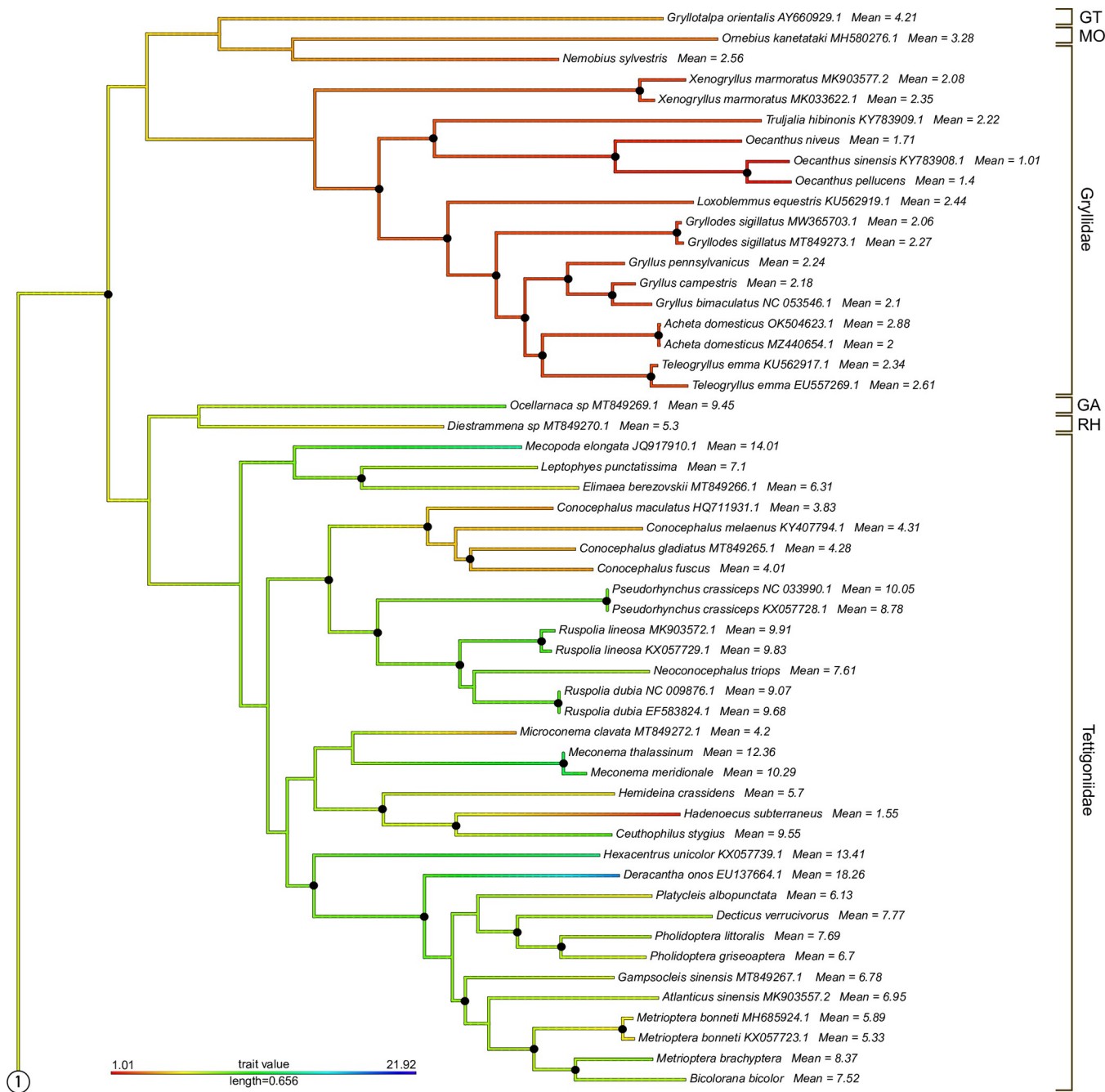

**Fig 2. Maximum likelihood phylogenetic tree of the mitochondrial dataset of Orthoptera.** Black dots on nodes represent Bootstrap support values of >95%. Branch colors code genome size (see legend). Mean genome size values are given for all tips. The circled connector "1" links this tree to the tree in Fig 3. Abbreviations for higher taxa: GT = Gryllotalpidae, MO = Mogoplistidae, GA = Gryllacrididae, RH = Rhaphidophoridae.

group to Meconematinae, albeit with poor support of 70% Bootstrap (S1 File). Note that *Hemideina* measurements were generated with the DAPI stain, which is influenced by AT/GC ratio in the genome and therefore considered less accurate than the PI stain used in all other FCM values listed here. *Acrida* was retrieved as the sister group to all other Acrididae, which contrasts with previous hypotheses [37, 40].

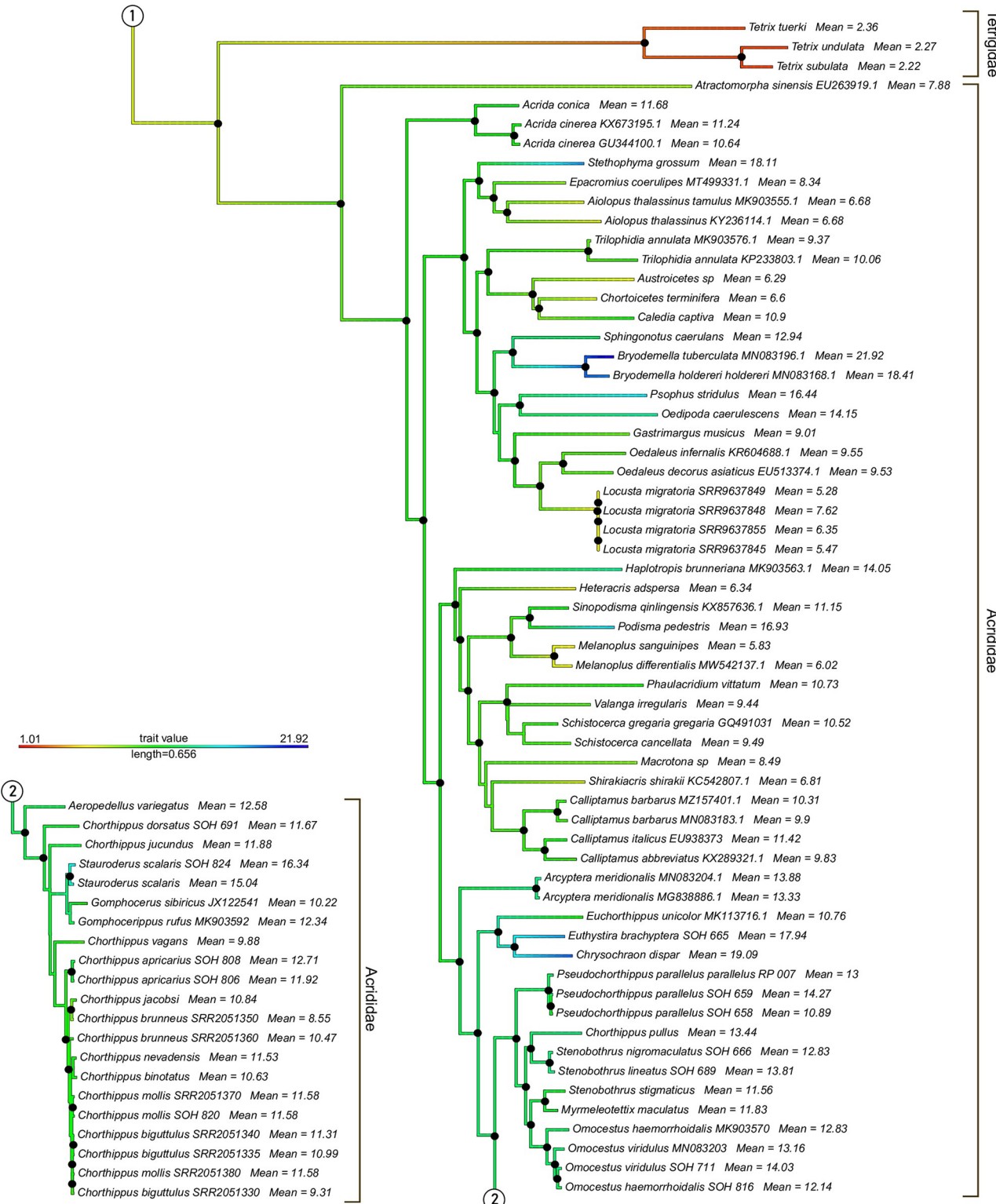

**Fig 3. Maximum likelihood phylogenetic tree of the mitochondrial dataset of Orthoptera.** Black dots on nodes represent Bootstrap support values of >95%. Branch colors code genome size (see legend). Mean genome size values are given for all tips. The circled connector "1" links this tree to the tree in Fig 2 The circled connector "2" links parts of the tree shown in this figure.

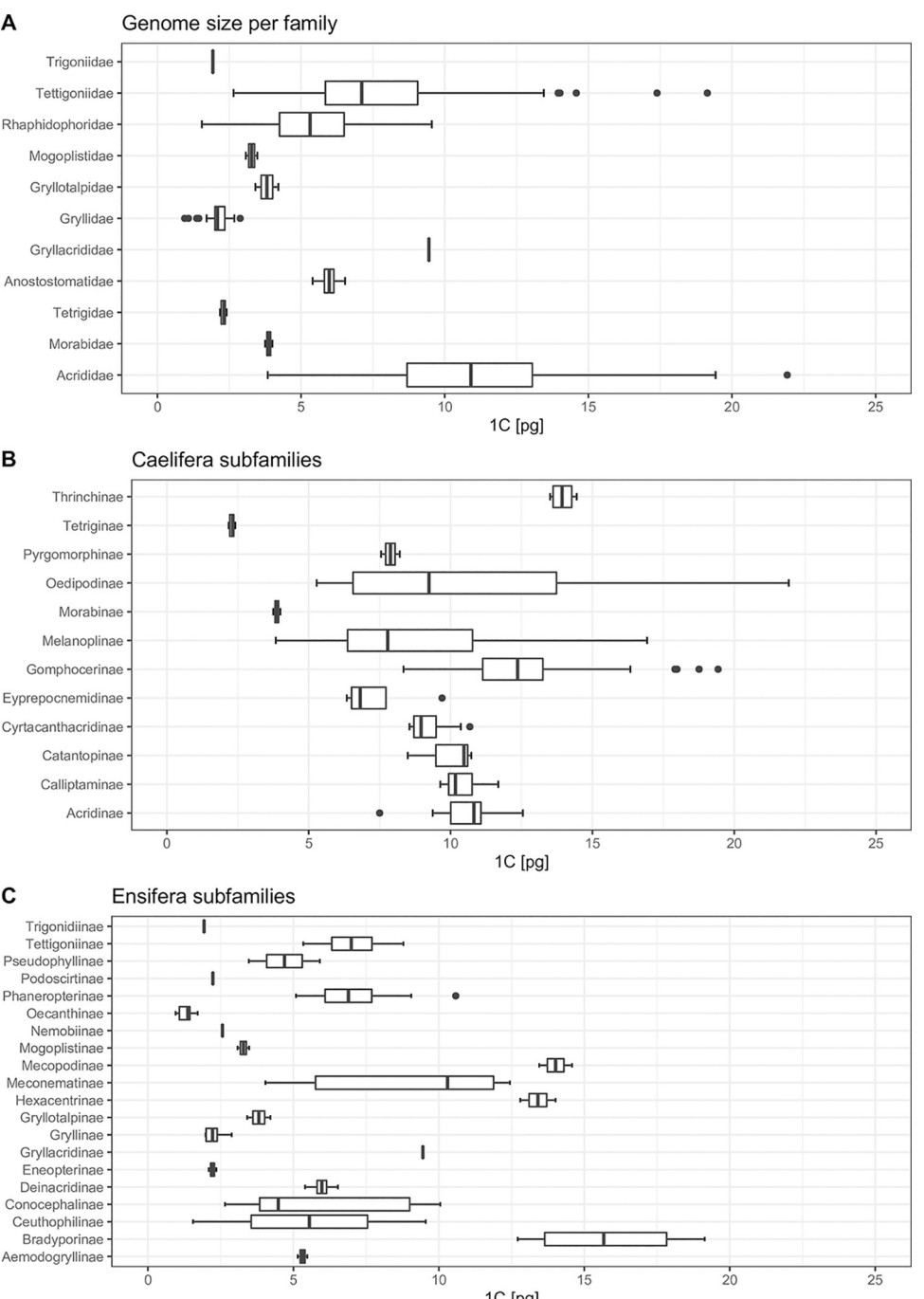

**Fig 4.** Box plots of genome size in families of Orthoptera (A), in subfamilies of Caelifera (B) and Ensifera (C).

The reconstructed paraphyly of genera *Ruspolia* and *Metrioptera* most likely reflects the urgent need of revising their taxonomy. The genus *Chorthippus* is notoriously complicated, and a revision will have to be based on more comprehensive data, as mitochondrial datasets have been shown to yield phylogenetic results incongruent to those based on larger genomic datasets [42].

Our phylogenetic tree also shows that exceptionally large genome sizes (more than 1C = 16 pg) are attained only in isolated clades. This result is certainly restricted by our dataset, which includes only few representatives of many clades. Nevertheless, it shows that large genome sizes are characteristic of only single genera (*Bryodemella*, *Deracantha*, *Stethophyma*) or closely related genera, e.g.. *Chrysochraon* + *Euthystira*. Husemann et al. [43] estimated the split of *Bryodemella* from *Sphingonotus* to about 31.8 million years ago (ma) [38]. The split of *Euthystira* from *Euchorthippus* was estimated to 12.88 [15.69–9.96] ma by Hawlitschek et al. [42]. There are no estimates for the split of *Stethophyma* from *Epacromius* + *Aiolopus*. Estimating the age of the lineage of *Deracantha* is difficult due to its contradictory placing in phylogenetic trees by Mugleston et al. [36] and Yuan et al. [12], but lineages in Ensifera are generally older than in Caelifera. The splits from related clades with smaller genome sizes (e.g., *Phaneroptera*) have been dated to around 100 ma in these studies. Based on these estimates, some large genomes may be of comparatively old evolutionary age. However, the increase in genome size is not necessarily related to the splitting of lineages we were able to detect. Duplications of chromosomes may have played a role for speciation with subsequent merging of chromosomes to the original number, but due to lack of evidence this remains speculation. Much finer phylogenetic resolution at the genus and species level will be required to track the evolution of genome size in individual clades more reliably.

### The relationship of genome size with life history and cytogenetic traits

No previous studies have been able to answer the question as to why some species of grasshoppers have such large genomes. Some of the earlier record keepers, *Podisma pedestris* and *Stauroderus scalaris*, as well as *Bryodemella tuberculata*, are species of montane habitats in Central Europe today. However, this does not hold true for their current global and historical European distributions [15]. Our sampling for this study was restricted to central Europe, but more species of other regions need to be studied to detect possible correlations between genome size and ecological or geographic variables.

Hypotheses on a correlation between life history traits and genome size have also been raised, e.g., body size and the ability to fly [44–46]. Larger body size was hypothesized to correlate with larger genomes, whereas the genomes of flying species were hypothesized to be smaller than those of flightless species. Yuan et al. [12] tested for any correlation of these traits with genome size in their dataset of tettigoniid ensiferans but found none. We did not test this in our dataset because it covers a wide phylogenetic range at rather coarse taxonomic resolution, and a finer scale will probably be necessary to detect any such correlation. Caeliferan species with particularly large genomes, such as *Chrysochraon dispar* and *Podisma pedestris* are flightless (with the exception of rare long-winged individuals), whereas *Bryodemella tuberculata* and *Stethophyma grossum* are good fliers [15]. Among the Ensifera, *Deracantha onos* is large-bodied and flightless, as is *Hemideina crassidens*, whose genome is substantially smaller (1C = 5.7 pg). No other large flightless ensiferan genome has been analyzed, which makes the search for correlation between these traits and genome sized difficult.

Our dataset includes two cave-dwelling species, *Ceuthophilus stygius* and *Hadenoecus subterraneus* (both Gryllidea–Rhaphidophoridae–Ceuthophilinae). While both species have very similar chromosome counts (36+X0 vs. 34+X0), the genome of *C. stygius* is almost the five-fold size of that of *H. subterraneus* (9.55 pg vs. 1.55 pg). It is therefore difficult to speculate if the adaptation to the cave environment has any consequences for genome size. Notably, Gryllidea are overall very heterogeneous regarding genome size (0.95 pg to 9.55 pg) and chromosome count (10+X0 to 36+X0).

Other than life history and ecology, genome size has been hypothesized to correlate with traits of cytogenetics and genome architecture. Several studies found large genomes,

including those of orthopterans, rich in satellite DNA, long terminal repeats and transposons, including helitrons and mariner like elements [9, 10, 47–49]. Whole genome duplications are another presumably common reason for large genome size (e.g., in fish [50]), going along with a polyploidization and a large number of chromosomes. However, such a relation was not found in Orthoptera. Intuitively, taxa with more chromosomes might be expected to also have larger genome sizes. Our analysis does not suggest any positive correlation of chromosome number and genome size. Some taxa with especially small numbers of chromosomes, such as European Gomphocerinae, have some of the largest genomes [8]. This suggests that the chromosome number reduction is associated with fusions rather than the actual loss of chromosomes.

On the other hand, we found, despite an overall rather narrow range of GC content, a correlation between genome size and GC content. Larger genomes had a generally higher GC content. There was no indication of difference in GC content between sexes, but the families studied here differed significantly. Acrididae and Tettigoniidae (with large genomes) were found to have genomes with higher GC content compared to Tetrigidae and Gryllidae (with small genomes). The general implications of GC content on animal genomes are not well studied. Low GC content may be an indicator of the presence of bacterial endosymbionts [51], whereas high GC content may be a sign of low chromatin condensation [52]. How these phenomena affect insects has not been studied [53].

As the majority of Orthoptera investigated, most species included in our study follow an XX/X0 sex determination system, implying that female genomes should be larger than male genomes just due to the second copy of the sex chromosome X. The same can be assumed for species with XX/XY (here in *Oecanthus*), as neo-Y chromosomes should be smaller than X chromosomes [54]. We find this reflected in the difference between female and male genomes of most species. However, the differences are minuscule in some species and even inverted (with male genomes larger than female genomes) in *Chorthippus dorsatus* and *Myrmeleotettix maculatus* (both Acrididae). In the case of *M. maculatus*, the inversion can most likely be attributed to different methods used to measure genome size (Feulgen densitometry vs. Flow cytometry) of males and females in different studies. Conversely, the specimens of *Ch. dorsatus* were from the same locality and measured in the same workflow, suggesting real intraspecific variability. The presence of B chromosomes might offer an explanation for the larger genome size of males than females, but no such phenomena have been reported for this species [55, 56] and chromosomes of specimens were not analyzed in the present study. Conclusions drawn on intraspecific difference in genome size will have to be backed by much larger sample sizes, but we uphold that the comparison of our present genome size measurements with the same species reported by previous studies show sufficient overall congruence to allow for interspecific comparisons and the tracking of genome size evolution.

Finally, genomic sequence data will be necessary to investigate the reasons behind the huge genomes of Orthoptera. Currently, better transcriptome and genome assemblies are on the way which may help to better understand the reasons for the large sizes of Orthoptera genomes [35, 57].

## Supporting information

**S1 Table. Complete table of all genome size measurements of Orthoptera reviewed for this study.**
(XLSX)

**S1 File. The maximum likelihood phylogenetic tree reconstructed in IQtree.**
(TRE)

## Acknowledgments

We thank the Government of Upper Bavaria for issuing permits for the collection of specimens of protected species. This work benefited from expertise sharing and discussions within the DFG priority program SPP 1991.

## Author Contributions

**Conceptualization:** Oliver Hawlitschek, David Sadílek, Pavel Trávníček, Martin Husemann.

**Data curation:** Oliver Hawlitschek, Lara-Sophie Dey, Katharina Buchholz, Sajad Noori, Timo Wehrt, Jason Brozio, Matthias Seidel, Martin Husemann.

**Formal analysis:** Oliver Hawlitschek, David Sadílek, Lara-Sophie Dey, Katharina Buchholz.

**Investigation:** Oliver Hawlitschek, David Sadílek, Lara-Sophie Dey, Katharina Buchholz, Sajad Noori, Inci Livia Baez, Timo Wehrt, Jason Brozio, Pavel Trávníček, Matthias Seidel, Martin Husemann.

**Methodology:** Pavel Trávníček.

**Project administration:** Oliver Hawlitschek, Martin Husemann.

**Resources:** Martin Husemann.

**Supervision:** Oliver Hawlitschek, Martin Husemann.

**Validation:** Oliver Hawlitschek, David Sadílek, Inci Livia Baez, Martin Husemann.

**Visualization:** Oliver Hawlitschek.

**Writing – original draft:** Oliver Hawlitschek, David Sadílek, Martin Husemann.

**Writing – review & editing:** Oliver Hawlitschek, David Sadílek, Lara-Sophie Dey, Katharina Buchholz, Sajad Noori, Inci Livia Baez, Timo Wehrt, Jason Brozio, Pavel Trávníček, Matthias Seidel, Martin Husemann.

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
