## [Decision Letter · Decision Letter 0]

20 Jan 2023

PONE-D-22-25859

New estimates of genome size in Orthoptera and their evolutionary implications

PLOS ONE

Dear Dr. Hawlitschek,

Thank you for submitting your manuscript to PLOS ONE. After careful consideration, we feel that it has merit but does not fully meet PLOS ONE’s publication criteria as it currently stands. Therefore, we invite you to submit a revised version of the manuscript that addresses the points raised during the review process.

I have received evaluations from 2 reviewers. They raised some positive aspects of the manuscript, mainly related to study design and explanations for differences in genome sizes.

However, a reviewer indicated some points that need your attention, amendment and improvement, especially in respect to the phylogenetic and comparative analysis. I agree with him. Please clarify the issues raised by the reviewers and check their suggestions.

Please, I would like you to know that the final acceptance of your manuscript will depend on the quality of the review of your manuscript and the responses to the reviewers' comments. Please let me know if you have any questions.

We look forward to receiving your revised manuscript.

Kind regards,

Maykon Passos Cristiano, D. Sc.

Academic Editor

PLOS ONE

and https://journals.plos.org/plosone/s/file?id=ba62/PLOSOne_formatting_sample_title_authors_affiliations.pdf.

Additional Editor Comments:

In addition, I would like to make some suggestions:

Line 103: the word “sacrifice” refers to religious actions, please change it.

Line 110: Why did the authors use the Pisum sativum as an internal standard rather than an insect? Like Drosophila melanogaster or Tenebrio?

Line 125: What formula was used to calculate genome sizes?

Reviewers' comments:

Reviewer's Responses to Questions

**Comments to the Author**

1. Is the manuscript technically sound, and do the data support the conclusions?

Reviewer #1: Partly

Reviewer #2: Yes

2. Has the statistical analysis been performed appropriately and rigorously? 

Reviewer #1: Yes

Reviewer #2: Yes

3. Have the authors made all data underlying the findings in their manuscript fully available?

Reviewer #1: Yes

Reviewer #2: Yes

4. Is the manuscript presented in an intelligible fashion and written in standard English?

Reviewer #1: Yes

Reviewer #2: Yes

5. Review Comments to the Author

Reviewer #1: This study is about variation of genome size across the order Orthoptera, an insect order where genomes are known to vary considerably in size. The study uses existing resources and data and presents new genome size estimates for 50 species of Ensifera and Caelifera, which represents a great addition to the existing data. The authors found a new largest measured genome for all insects in the species Bryodemella tuberculata (Caelifera) and they used these estimates to reconstruct ancestral genome sizes based on a phylogenetic tree reconstructed using mitochondrial genomic data.

The study is well presented and written, and both the new data and the conclusions are worth publication. The discussion in itself, reviewing the different causes tested or invoked to explain differences in genome sizes, is very interesting.

I have a few issues about the phylogenetic aspects of the study though, in terms of methodology:

1) The phylogenetic analysis is quite basic, but ML with IQ tree is usually reliable, which makes it acceptable as a reference, but this is not surprising that the tree is not completely resolved. It would have been better to reconstruct the phylogeny using missing data for some species, but keeping the whole mitogenomes when avaiblable in order to maximize the informative content of the data set. This may not change drasticaly the results, which are coherent with previous phylogenies, but it could solve an few uncertainties mentionned by the authors.

2) One important detail is missing in the materials and method section: did the authors partitioned their data set by gene (recommended), or did they consider only one partition for the whole set of genes?

3) There is potentially a problem in the study about the comparative methods used to infer ancestral states: the tree from the phylogenetic analysis usually have to be ultrametric to be used for ancestral state reconstructions, which is not the case with trees coming directly from IQtree. A dating analysis or a step of ultrametrisation has be be added in order to respect this principle in phylogenetic comparative analyses.

4) Another problem in the manuscript is the absence of detailed methods explaining the methods used to infer ancestral states, and why some species present two values in the phylogeny for a given species. Citing the packages is not enough, there should be details about how analyseses were made (which functions and options) and a could be script given in supplemntary material ideally.

Reviewer #2: Review of Manuscript: PONE-D-22-25859, Hawlitschek et al 2023

General comments on the manuscript:

The authors present a well designed study with clearly stated goals and these goals are thoroughly explored throughout the manuscript. Additionally, I strongly recommend to also submit the genome size estimates to Animal Genome Size Database.

Lines 158-161:

These lines suggest that the authors coded an analysis in the R environment using ‘ape’ and ‘phytools’ packages. It would be ideal if the code and data could be deposited in appropriate data repositories (for example see: data dryrad, https://datadryad.org/). Furthermore, other data and code generated from and during this study can also be archived.

Minor edits:

Line 139: “As this test resulted non-significant (Results),” should perhaps be: “As this resulted in non-significant results,”

Recommendation:

I recommend to publish this study given the minor edits and suggestions are implemented.

6. PLOS authors have the option to publish the peer review history of their article (what does this mean?). If published, this will include your full peer review and any attached files.

Reviewer #1: No

Reviewer #2: No

---

## [Decision Letter · Decision Letter 1]

24 Feb 2023

New estimates of genome size in Orthoptera and their evolutionary implications

PONE-D-22-25859R1

Dear Dr. Hawlitschek,

We’re pleased to inform you that your manuscript has been judged scientifically suitable for publication and will be formally accepted for publication once it meets all outstanding technical requirements.

Kind regards,

Michael Schubert

Academic Editor

PLOS ONE

Reviewers' comments:

Reviewer's Responses to Questions

**Comments to the Author**

1. If the authors have adequately addressed your comments raised in a previous round of review and you feel that this manuscript is now acceptable for publication, you may indicate that here to bypass the “Comments to the Author” section, enter your conflict of interest statement in the “Confidential to Editor” section, and submit your "Accept" recommendation.

Reviewer #2: All comments have been addressed

2. Is the manuscript technically sound, and do the data support the conclusions?

Reviewer #2: Yes

3. Has the statistical analysis been performed appropriately and rigorously? 

Reviewer #2: Yes

4. Have the authors made all data underlying the findings in their manuscript fully available?

Reviewer #2: Yes

5. Is the manuscript presented in an intelligible fashion and written in standard English?

Reviewer #2: Yes

6. Review Comments to the Author

Reviewer #2: (No Response)

7. PLOS authors have the option to publish the peer review history of their article (what does this mean?). If published, this will include your full peer review and any attached files.

Reviewer #2: No

---

## [Editor Report · Acceptance letter]

6 Mar 2023

PONE-D-22-25859R1 

New estimates of genome size in Orthoptera and their evolutionary implications 

Dear Dr. Hawlitschek:

I'm pleased to inform you that your manuscript has been deemed suitable for publication in PLOS ONE. Congratulations! Your manuscript is now with our production department. 

Kind regards, 

on behalf of

Dr. Michael Schubert 

Academic Editor

PLOS ONE